# Polyamine Signal through HCC Microenvironment: A Key Regulator of Mitochondrial Preservation and Turnover in TAMs

**DOI:** 10.3390/ijms25020996

**Published:** 2024-01-13

**Authors:** Qingqing Liu, Xiaoyu Yan, Runyuan Li, Yuan Yuan, Jian Wang, Yuanxin Zhao, Jiaying Fu, Jing Su

**Affiliations:** Key Laboratory of Pathobiology, Department of Pathophysiology, Ministry of Education, College of Basical Medical Sciences, Jilin University, 126 Xinmin Street, Changchun 130012, China; lqq22@mails.jlu.edu.cn (Q.L.); yanxy@jlu.edu.cn (X.Y.); lry23@mails.jlu.edu.cn (R.L.); yuany23@mails.jlu.edu.cn (Y.Y.); wjian21@mails.jlu.edu.cn (J.W.); yuanxinz22@mails.jlu.edu.cn (Y.Z.); fujy21@mails.jlu.edu.cn (J.F.)

**Keywords:** TME, polyamine, mitochondria, spermidine, EVs

## Abstract

Hepatocellular carcinoma (HCC) is the most common primary liver cancer, and, with increasing research on the tumor immune microenvironment (TIME), the immunosuppressive micro-environment of HCC hampers further application of immunotherapy, even though immunotherapy can provide survival benefits to patients with advanced liver cancer. Current studies suggest that polyamine metabolism is not only a key metabolic pathway for the formation of immunosuppressive phenotypes in tumor-associated macrophages (TAMs), but it is also profoundly involved in mitochondrial quality control signaling and the energy metabolism regulation process, so it is particularly important to further investigate the role of polyamine metabolism in the tumor microenvironment (TME). In this review, by summarizing the current research progress of key enzymes and substrates of the polyamine metabolic pathway in regulating TAMs and T cells, we propose that polyamine biosynthesis can intervene in the process of mitochondrial energy metabolism by affecting mitochondrial autophagy, which, in turn, regulates macrophage polarization and T cell differentiation. Polyamine metabolism may be a key target for the interactive dialog between HCC cells and immune cells such as TAMs, so interfering with polyamine metabolism may become an important entry point to break intercellular communication, providing new research space for developing polyamine metabolism-based therapy for HCC.

## 1. Introduction

Immune escape is an important feature that occurs during the development of HCC [1]. In HCC cancer tissues, sparse immune cell infiltration has led to “immune-cold” cancer as an important feature of HCC. HCC cell growth is associated with immunosuppression, and immunosuppressive cells (regulatory T cells and M2-TAMs) play an important role in cold cancers. M2-TAMs contribute to the formation of an immunosuppressive HCC microenvironment by the up-regulation of arginase 1 (ARG1) and IL-10, which leads to T cell depletion and NK cell inactivation [2,3], thus influencing the progression of HCC. Recent studies have shown that cellular metabolism plays a critical role in supporting the maintenance and development of immune cells and is a dominant force in immune regulation [4]. Mitochondria, as cellular energy metabolism centers and energy factories, are involved in the pathogenesis of HCC by synthesizing adenosine triphosphate (ATP) through phosphorylation (OXPHOS), which plays an important role in HCC tumorigenesis, patient survival, and tumor progression [5]. As for immune cells, among immune-activated cells such as Th17 [6] and M1-type macrophages [7], they mainly rely on glycolysis for energy supply and play an anti-tumor role. In contrast, in immunosuppressive cells such as regulatory T cells or M2-type macrophages, they are mainly dependent on oxidative phosphorylation (OXPHOS) for energy supply and exert pro-tumorigenic effects, suggesting that immune function is closely related to the mode of mitochondrial energy metabolism.

Polyamines have pro-tumorigenic and immunosuppressive properties. Arginine promotes polyamine metabolism, and ARG1 metabolizes arginine to ornithine, which supplies polyamine synthesis, including spermidine (SPD) production [8]. It was found that tumor-associated DCs deplete arginine and limit its use in T cells, and DCs metabolize arginine to SPD, further driving tryptophan metabolic enzyme indoleamine-2,3-dioxygenase (IDO) phosphorylation and IDO signaling activation [9]. This contrasts with the production of SPD in T cells, which enhances their anti-tumor effects [10]. IDO catabolizes tryptophan to kynurenine, whose expression is increased in cancer cells and activated immune cells [11]. This illustrates the overlapping needs between cancer cells and immune cells. T cells require tryptophan for proliferation and activation [12]. Targeting cancer cells with IDO inhibitors may suppress T cell-mediated immunity by limiting tryptophan metabolism. Overexpression of IDO in tumor cells impairs T cell responses, possibly by driving tryptophan degradation in cancer cells and limiting tryptophan supply to T cells [13]. Overall, these roles of DCs limit the anti-tumor immunity of T cells. Polyamines play different roles in cancer cells and immune cells. Increased polyamine production not only maintains continued proliferation of cancer cells, but it also shifts the energy metabolism of immune cells towards aerobic respiration, promoting enhanced mitochondrial function, mediating T cell differentiation and macrophage polarization, and conferring anti-inflammatory properties on T cells and macrophages. Puleston et al. found that polyamine biosynthesis results in hypusinated eukaryotic translation initiation factor 5A (eIF5A^H^) promoting the efficient expression of a subset of mitochondrial proteins involved in the TCA cycle and OXPHOS, which promoted macrophage M2 activation [14]. Polyamine metabolism controls CD4 helper T cells differentiation into distinct functional fates [15]. In addition, Tregs have altered metabolism compared to conventional T cells. Berod et al. showed that Foxp3(+) regulatory T cells isolated directly from mice showed increased fatty acid-driven oxidative phosphorylation (OXPHOS) [16]. So, does polyamine metabolism in tumor cells and immune cells interact and influence each other?

Studies have shown that HCC is enriched in exosomes and that tumor-derived exosomes contribute to the shaping of the immunosuppressive TME by influencing the communication between cancer cells and immune cells. Extracellular vesicles (EVs), natural membrane-encapsulated nanoparticles secreted by all cells, are important mediators of intercellular communication and are divided into three main categories: exosomes, micro-vesicles, and apoptotic vesicles [17,18,19]. EVs are the key form of intercellular transmissions of polyamines, which are extracellular lipid bilayer micro-vesicles that regulate physiological and pathological processes in the body and are secreted by cells containing proteins, nucleic acids, amino acids, and other metabolites, suggesting that the intercellular transmitting role of polyamines is important. Studies have shown that HCC-derived exosomes play a crucial role in promoting intercellular communication and signaling within the HCC TME [20,21]. However, the specific role played by polyamine metabolism in the communication between the two is currently unclear, and thus understanding the interaction between tumors and immune cells could further contribute to the development of cancer therapy.

It was shown above that mitochondrial OXPHOS capacity plays a role in regulating immune cell phenotypes in immune cells, and the process of mitochondrial quality control maintains stable energy metabolism levels by repairing or eliminating “unhealthy mitochondria” at the molecular, organelle, or cellular levels. Mitochondrial quality control is maintained by mitochondrial autophagy, a process that selectively removes damaged or excess mitochondria through the autophagic mechanism, thereby fine-tuning mitochondrial number and maintaining energy metabolism. In addition, autophagy is a fundamental mechanism for the provision of OXPHOS nutrients, which are essential for tumor maintenance and progression [22]. Therefore, mitochondrial autophagy is a possible mechanism for modulating immune phenotypes. Furthermore, it was found that mitochondria control autophagy by regulating the mitochondria-endoplasmic reticulum contact site, which promotes increased OXPHOS levels [23]. Importantly, polyamines promote mitochondrial autophagy to inhibit inflammation. Eisenberg et al. showed that supplementation with SPD enhanced cardiac autophagy, mitochondrial autophagy, and mitochondrial respiration and also improved the mechano-elastic properties of cardiomyocytes in vivo to inhibit inflammatory progression [24]. The above suggests that the mechanism by which polyamines upregulate OXPHOS to promote M2-type macrophage polarization may be related to mitochondrial autophagy.

This paper summarizes the potential mechanisms by which polyamine metabolism influences the remodeling of the TIME by regulating the level of energy metabolism of immune cells, as well as the potential for the transfer of polyamines between tumors and immune cells via EVs. The remainder of this paper also explores the prospect of targeting polyamines for antitumor therapy, providing new research space for the development of polyamine metabolism-based therapies for HCC.

## 2. Polyamine Metabolism and HCC

### 2.1. Polyamine Metabolism Promotes the Progress of HCC

Polyamines, mainly including putrescine, SPD, and spermine (SPM), are produced by decarboxylation of ornithine by ornithine decarboxylase (ODC), and the product of this reaction putrescine is converted to SPD and SPM by the gradual addition of a propylamine group by the spermidine synthase (SRM) and spermine synthase (SMS), respectively. Yue et al. showed that spermidine-treated cells exhibited increased microtubule-associated protein stability and autophagic flux, as well as attenuated chemical damage-induced liver fibrosis and HCC foci in an in vivo assay, which ultimately significantly prolonged the lifespan of experimental mice [25]. Interestingly, S-adenosylmethionine decarboxylase proenzyme (AMD1) is a key enzyme involved in the synthesis of SPD, and AMD1 up-regulates the stemness of hepatocellular carcinoma cells through obesity-associated protein (FTO)-mediated mRNA demethylation; a poor prognosis is suggested when AMD1 is enriched in human HCC tissues [26]. This suggests that polyamine levels assume different roles in normal and tumor cells.

Indeed, cancer cells require continuously elevated intracellular polyamine pools to maintain their ability to sustain proliferation [27,28]. Elevated polyamine levels in cancer cells are maintained by increased biosynthesis, increased transport, and decreased catabolism, and studies have shown that oncogenes are important regulatory signals for polyamine metabolism levels within cancer cells. Many oncogenes, including *MYC* and *BRAF*, contribute to increased polyamine levels in cancer cells (Figure 1). Bello-Fernandez et al. demonstrated that *ODC1* is a transcriptional target of the *MYC* oncogene in fibroblasts and that increased expression of *MYC* leads to an increase in *ODC1* mRNA, which promotes an increase in the amount of polyamines required for cell division and proliferation [27]. In contrast, ammonia accumulation due to inhibition of the urea cycle metabolic pathway by the oncogene Tp53 can directly down-regulate *ODC1* mRNA translation, which, in turn, leads to a decrease in overall intracellular ODC activity, resulting in the blockage of polyamine synthesis and slowing down tumor cell proliferation [29]. Therefore, polyamine metabolism in HCC can evaluate the extent of malignancy [30]. Furthermore, the RAS-RAF-MEK-ERK signaling pathway has been shown to regulate polyamine metabolism in several ways. Activated RAS increases polyamine uptake by human colon cancer cells by regulating caveolar endocytosis [31]. Thus, polyamine metabolism often plays a key role in tumor progression.

Elevated polyamine levels are commonly observed in immune cells in the TME and are required for the progression of various types of cancers. The TME refers to the complex network of cells and molecules that surround and support tumor growth [32], and it is usually immunosuppressive, which allows cancer cells to evade immune surveillance [33]. Increased levels of polyamines can induce the formation of an immunosuppressive microenvironment. Chia et al. showed that the blockade of polyamine metabolism in cancer resulted in reduced tumor growth, the effects of which may be mediated by increased T cells infiltration and a macrophage pro-inflammatory phenotype [34]. Studies have shown that SPD promotes macrophage polarization toward an anti-inflammatory phenotype [35,36]. Indeed, macrophage arginine variability determines the polarization phenotype, with M1-type macrophage polarization relying on nitric oxide synthase for the conversion of arginine to NO. In contrast, M2-type macrophage polarization relies on ARG1 for the metabolism of arginine to ornithine. Ornithine then enters the polyamine metabolic cycle, where intermediate metabolites such as SPD energize other metabolic pathways in M2-type macrophages [37,38]. The immunomodulatory properties of SPD are largely attributed to the hypothesis that SPD induces autophagy or SPD-dependent eIF5A^H^. SPD is the sole substrate for eIF5A^H^, which is essential for protein translation [39]; Zeng et al. showed that reducing eIF5A^H^ inhibited cancer cell growth and polarization of M2-TAMs in oral cancer [40]. Similarly, polyamine blockade therapy (PBT) co-treatment with DFMO and trimeric polyamine transport inhibitor (PTI) decreased M2-TAMs responses and increased M1-TAMs immunoreactivity [41].

Polyamine metabolism is essential for T cell differentiation and contributes to the maintenance of their functional stability. Polyamine metabolism is primarily responsible for the ability of helper CD4^+^ T cells to differentiate into their functional subpopulations, including helper T cell 1 (TH1), TH2 and TH17, and Treg cells, through interactions with the epigenome and the tricarboxylic acid cycle. Inhibition of polyamine synthesis by ODC1 deficiency, leading to a reduction in eIF5A^H^, induces an irregular anti-cancer T cell phenotype [15,42]. Guilhermina et al. demonstrated that the immunomodulatory effect of SPD in T cells in the T cell transfer-induced colitis model is mediated by autophagy and that in vitro and in vivo it is possible to alter the equilibrium of the differentiation of CD4^+^ T cells from Th17 to Foxp3-expressing Treg cells to enhance their immunosuppressive function; activated T cells lacking the autophagy gene Atg5 cells cannot undergo this change [43]. Arginine, as a major donor for polyamine synthesis by T cells, is essential for efficient T cell activation and function. Increased depletion of arginine by tumor cells and M2-TAMs reduces its availability to support cytotoxic T cells proliferation and function [44]. Similarly, Munder et al. showed that ARG1-containing cellular exosomes inhibit ovarian cancer T cell immune responses and promote ovarian cancer progression [45].

Thus, on the one hand, cancer cells require a continuously elevated intracellular pool of polyamines to maintain proliferation, and, on the other hand, polyamines promote the formation of an immunosuppressive TME to participate in tumor progression.

### 2.2. Involvement of Polyamines in Communication between HCC Cell and Immune Cells

Exosomes belong to a class of EVs between 30 and 100 nm in diameter with a lipid bilayer and are secreted extracellularly by a variety of cells, where they play a role in immune responses, cellular communication, and intercellular regulation, and where they carry proteins, nucleic acids, and lipids that affect the activity of recipient cells [20]. Current research suggests that tumor-derived exosomes can function directly in immune cells and that the specific role is determined by the exosome contents. MiR-146a-5p in HCC-derived exosomes was found to remodel macrophages and leads to M2-TAMS polarization through activation of NF-κB signaling and induction of pro-inflammatory factors [46]. circGSE1 in HCC-derived exosomes, through regulation of the miR-324-5p /TGFBR1/Smad3 axis, induce Treg cell expansion to promote HCC progression [47]. In addition, MicroRNA-15a-5p (miR-15a-5p) in HCC-derived exosomes was secreted into CD8^+^ T cells, and targeted binding directly inhibited PD1 expression and promoted HCC progression [48]. We therefore speculate that HCC cells can exert pro-tumorigenic functions by secreting polyamine-containing exosomes that act on immune cells.

The dynamic homeostasis of polyamines is a tightly regulated process. As discussed above, tumor cells maintain their sustained proliferative capacity by consistently elevating the intracellular polyamine pool, and aberrant increases in polyamine metabolism in immune cells promote the shaping of an immunosuppressive TME. Therefore, it is worth exploring whether there is a potential link between the polyamine metabolism of tumor cells and immune cells. Studies have shown that vesicles are a key pathway for polyamine transport between cells and that the Vesicular polyamine transporter (VPAT) is responsible for the vesicular storage and release of polyamines [49]. Polyamine secretion is dependent on the involvement of multiple VPATs. In humans, seven types of VPAT, including those with organic cation transporter 2 and multidrug and toxic compound extrusion 1, have been reported to transporter polyamines [50]. In Plasmodium falciparum, the Plasmodium falciparum CQ resistance transporter is a H(+)-coupled polyspecific nutrient transporter protein that transports polyamines [51]. Most polyamine transporter proteins known to date are localized to the plasma membrane and play a role in the cellular secretion of polyamine vesicles [52]. The current study shows that SPM in endothelial cell-derived exocytotic vesicles mediates smoking-induced pulmonary hypertension via calcium-sensitive receptors [53]. In addition, Puhka et al. analyzed metabolite enrichment of ex-vesicles in body fluids and found enrichment of products such as SPD and arginine [54], demonstrating that polyamines can be secreted extracellularly in the form of vesicles.

Current models for mammalian polyamine uptake include the following two main factors: (1) Polyamines enter the cell via polyamine permeases, which then translocate free cytoplasmic polyamines into polyamine-sequestering vesicles (PSVs). (2) receptor-mediated endocytosis, which probably involves glypian1 in caveolin 1-rich membrane regions [55]. These models suggest that polyamine translocation to intracellular stores in PSVs, from which they can be released into the cytoplasm, plays a role. In addition, it was shown that ATP13A2 function influences total intracellular polyamine concentration by promoting polyamine endocytosis and polyamine release into the cytoplasmic lysate [56], which is consistent with the receptor-mediated endocytosis model. The presence of extracellular “large amounts” of shared polyamines was demonstrated in T cells specific ODC-deficient mice, and, despite the inhibition of newly synthesized polyamines, such T cell levels of putrescine could be maintained by a compensatory increase in extracellular putrescine uptake [57]. Arginine is the primary carbon source for polyamine synthesis, and this extracellular pathway for obtaining polyamines may actually be more advantageous than the de novo synthesis pathway because it does not deplete the intracellular arginine pool for proliferating or differentiated T cells. Having mentioned above that immune cells can receive tumor-derived exosomes to interfere with their differentiation, we propose that immune cells can receive polyamines from tumor cell-derived exocysts, causing the immune cells’ polyamine pool to rise, which, in turn, affects the immune phenotype of their immune cells.

In summary, HCC-derived exosomes can modulate immune responses, serve as diagnostic and prognostic biomarkers, and provide new therapeutic targets for HCC [58,59]. Exosomes serve as one of the key pathways for signaling by proteins such as polyamines, and polyamine-rich exosomes of an HCC origin can directly enter immune cells and influence their function to exert their anti-tumor or pro-tumor effects. The TME dynamically evolves with cancer progression, and immune cells are constantly influenced by tumor cells, with polyamines in tumor-derived exosomes acting as a bridge between them. Targeting tumor-derived polyamines is, therefore, an important approach to overcome the formation of an immunosuppressive TME.

## 3. Polyamines Promote Immunosuppressive Microenvironment by Regulating Mitochondrial Energy Metabolism

Abnormal energy metabolism (Warburg effect) is one of the characteristics of tumor cells and plays a crucial role in tumorigenesis and survival. The blood supply of HCC cells is mainly from the hepatic artery, which provides sufficient oxygen for OXPHOS [60]. After the establishment of hepatitis C virus (HCV) replication, downregulation of key mitochondrial respiratory chain genes suggests that the virus promotes cancer cell-like metabolic reprogramming (“Warburg effect”). However, when cancer develops, increased OXPHOS levels can enhance the malignancy of HCC. Meanwhile, Tan et al. showed that the acquisition of host mtDNA by tumor cells lacking mtDNA enhances the respiratory function and tumorigenic potential of cancer cells [61]. The above demonstrates that mitochondrial respiration is required for tumorigenesis. In addition, differences in energy sources have become one of the important mechanisms of immune cells differentiation [62,63]. After hepatocarcinogenesis, immune cells are activated, and immune cells are supplied with energy in different ways. In pro-inflammatory cells, such as Th17 cells [64] and M1-type macrophages [65], energy is produced through increased glycolysis exerting an anti-tumorigenic effect, whereas, in regulatory cells, such as Treg cells or M2-type macrophages, energy is produced through increased mitochondrial function and β-oxidation exerting its pro-tumorigenic effect [66,67].

Numerous studies have shown that the SPD-eIF5A^H^ axis regulates mitochondrial OXPHOS function. Polyamine metabolism increases mitochondrial respiration by promoting mitochondrial respiration, and, in AML12 hepatic cells, SPD supplementation promotes increased expression of eIF5A^H^ and mitochondrial proteins and promotes OXPHOS production [68]. Similarly, Eisenberg et al. demonstrated that in mouse cardiomyocytes, dietary SPD induced mitochondrial autophagy, targeted damaged and potentially deleterious mitochondria, and improved mitochondrial abundance and respiration [69]. Importantly, the SPD-eIF5A^H^ axis regulates immune cell differentiation and development [42,70]. Puleston et al. showed that SPD serves as the sole substrate for hypusine modification of lysine residues in eIF5A and that increased SPD allows eIF5A^H^ to be enriched in macrophages [14]. eIF5A^H^ acts as a regulator of mitochondrial homeostasis to promote the efficient expression of a subpopulation of mitochondrial proteins involved in the TCA cycle and OXPHOS to drive polarization of M2-type macrophages [71]. In addition, Shi et al. showed that inhibition of ATP synthase significantly reduced Treg cell induction, demonstrating an OXPHOS-dependent Treg induction [72]. SPD, as a regulator of T cell differentiation and function, was shown to regulate CD4+ T cell differentiation in vitro, preferentially converting naïve T cells into a Treg cell phenotype to enhance their immunosuppressive function [43]. Therefore, the SPD-eIF5A^H^ axis may also play a role in promoting Treg cell value addition by increasing OXPHOS in Treg cells (Figure 2).

As mentioned above, the energy metabolism of immune cells is closely related to their activation state, whereas the potential mechanism by which polyamine metabolism mediates T cells differentiation remains elusive. In summary, polyamines promote the growth of cancer cells and modulate the differentiation of immune cells to promote the formation of an immunosuppressive microenvironment, which is closely related to the elevation of cellular mitochondrial OXPHOS levels.

## 4. Mitochondrial Autophagy Is an Important Pathway for the Enhancement of Mitochondrial Function by Polyamine Metabolism

Mitochondria are important organelles found in eukaryotic cells that play a key role in ATP production through OXPHOS. Autophagy is involved in the regulation of energy mediators and is a key regulator of cellular homeostasis and a potential target for tumor therapy. Current studies have shown that mitochondrial autophagy can be induced by SPD administration, which increases autophagy-dependent selective degradation of mitochondria in vivo in cardiomyocytes and in various cultured cell lines, thereby contributing to the maintenance of mitochondrial function [24,73,74].

Current studies suggest that polyamine metabolism promotes mitochondrial autophagy by modulating mitochondrial autophagy protein production and inducing mitochondrial depolarization. Eisenberg et al. demonstrated that SPD treatment markedly increased the positive areas of Mito-Keima in mouse cardiomyocytes, suggesting an increase in mitochondrial autophagy [24]. Qi et al. showed that SPD induces mitochondrial autophagy by triggering mitochondrial depolarization, which triggers the formation of mitochondrial autophagosomes and the fusion of mitochondrial lysosomes, and SPD induces the activation of the ataxia telangiectasia mutated protein (ATM), which promotes the accumulation of PINK1 and translocation of Parkin to the damaged mitochondria, ultimately leading to the reduction in mitochondrial numbers in GM00637 cells [74]. In addition, Hofer et al. demonstrated that elevated levels of dietary SPD affect mitochondrial function by enriching intracellular eIF5A^H^, which promotes an increase in ATG3 and ATG7, as well as PINK1-PRKN translation [75].

Growing evidence suggests that mitochondrial energy and metabolism are highly relevant to disease and cancer and that they not only perform critical bioenergetic and biosynthetic functions but also regulate ROS homeostasis and apoptosis. The contribution of mitochondrial autophagy in regulating metabolism may establish a balance between aerobic glycolysis and OXPHOS for cancer cell survival. In addition, mitochondrial autophagy may promote cancer cell plasticity through metabolic remodeling for better adaptation to the TME. Mitochondrial autophagy in immune cells serves as a bridge between polyamine metabolism and mitochondrial energy metabolism; thus, polyamine biosynthesis can regulate mitochondrial function through mitochondrial autophagy to affect macrophage polarization and T cell differentiation. Yao et al. showed that mitochondrial autophagy is frequently up-regulated in HCC cells with mitochondrial dysfunction and helps cancer cells to survive drug treatment or other stresses [76]. Li et al. revealed that PINK1-mediated mitochondrial autophagy promotes OXPHOS and redox homeostasis inducing a drug-tolerant persistent state in cancer cells enabling the cancer cells to escape from the cytotoxic stress brought about by anticancer therapy [77]. In addition, FBXL4 mutations cause excessive mitochondrial autophagy through BNIP3/BNIP3L accumulation, leading to mitochondrial DNA depletion syndrome (MTDPS), a group of genetic disorders characterized by a reduction in the number of copies of mitochondrial DNA, resulting in OXPHOS and mitochondrial functional defects [78]. Therefore, targeting mitochondrial autophagy in cancer cells may be an effective means of treating HCC.

## 5. Future Research Directions

### 5.1. Polyamines as Biomarkers for Cancer

As metabolomics technology continues to evolve and develop, one can more sensitively measure cancer metabolic profiles and identify biomarkers for cancer screening, diagnosis, and monitoring, suggesting new prospects for polyamines as cancer biomarkers and predictors of therapeutic response. Human epidermal growth factor receptor 2 metabolomics studies, also known as ERBB2-positive breast cancer, have shown that patients who responded well to trastuzumab-paclitaxel neoadjuvant therapy had higher levels of serum SPD than those who responded poorly, and lower levels of tryptophan [79]. Similarly, urinary and plasma polyamines and their metabolites are used diagnostically and as markers of tumor progression in lung and liver cancer [80,81]. Urinary N1 and N12-diacetyl spermine have been recognized as a valid biomarker for lung and ovarian cancer [82,83,84]. The combination of polyamine metabolomics analysis with genomic profiling could contribute to the development of more personalized cancer diagnostic and therapeutic approaches based on polyamines as biomarkers.

### 5.2. Polyamine Therapy

The high dependence of tumor cells on polyamines in HCC and the critical physiological role of polyamines in various immune cell types make targeting polyamine metabolic pathways a viable therapeutic strategy. Some of these strategies have been effective in preventing and treating cancer in animal models, and some have progressed to clinical trials. Although inhibitors have been designed for all polyamine biosynthetic enzymes, the most successful inhibitor to date has been DFMO, which irreversibly inhibits ODC through covalent binding to its active site, usually by depleting putrescine and SPD, leading to cellular arrest [28,85,86]. DFMO has also been evaluated as a potential co-cancer therapeutic agent. The in vivo effects of DFMO are strongest when combined with a polyamine-free diet, suggesting the utility of combining DFMO with polyamine transport inhibitors as a means of PBT [87,88]. In addition, while there are many ways to achieve immune reprogramming of cold tumors, strategies to reduce polyamine levels through PBT as an immunomodulator are a new but rapidly evolving area. A major benefit of PBT is that most cancers depend on elevated polyamine concentrations to sustain growth, and therefore a reduction in intracellular polyamine concentrations can inhibit cancer cell growth. Current approaches that can successfully target polyamine metabolism and its function include the following:(1)Inhibition of polyamine biosyntheses, such as ODC inhibitors, AdoMetDC inhibitors, SRM, and SMS inhibitors (Table 1).(2)Development of polyamine analogs such as BENSpm, PG-11047, CPENSpm, etc., unlike polyamine biosynthesis inhibitors, polyamine analogs accumulate, resulting in inhibition of polyamine biosynthesis pathways with concomitant increase in catabolic pathway activity [107] (Table 2).(3)Polyamine blockade therapy

Based on the characteristic increase in polyamine metabolism in tumor cells, PBT, a strategy that combines inhibition of polyamine biosynthesis with simultaneous blockade of polyamine transport (Table 3), may be more effective than therapies based on polyamine depletion alone and may involve an anti-tumor immune response. These findings open up new avenues of research for anticancer therapy utilizing aberrant polyamine metabolism [28]. The most exciting finding is that PBT promotes an antitumor immune response that produces even greater antitumor effects than would be expected from polyamine depletion in tumor cells alone. In an immunocompetent mouse model of lymphoma, melanoma, and colon cancer, the combination of DFMO with AMXT 1501 resulted in a decrease in tumor-infiltrating myeloid suppressor cells and an increase in CD3+ T cells, which inhibited tumor growth [119]. Furthermore, in the FVB mouse mammary tumor model, PBT protected mice from secondary tumor attack after primary tumor treatment and resection. In a similar study, a different polyamine transport inhibitor (Trimer PTI) was used in combination with DFMO. This resulted in a decrease in regulatory T cells and a decrease in tumor-infiltrating myeloid-derived suppressor cells, along with an increase in granzyme B and interferon-gamma (IFN-γ), and effector T-cell activation [41]. The end result was a suppression of the tumor-promoting microenvironment and an increase in the anti-tumor immune response.

The potential of PBT to enhance anti-tumor immune responses is consistent with recent studies using bone marrow-specific knockout ODC. ODC activity and polyamines favored tumor-resistant M2-type macrophage polarization while decreasing anti-tumor-resistant M1-type macrophage polarization. These findings are consistent with earlier studies on the involvement of ODC in the regulation of M1-type macrophages. In conclusion, the results of PBT and other studies suggest that interfering with polyamine metabolism would be a fruitful avenue for the development of anticancer therapies, possibly in conjunction with other immunotherapeutic modalities. PBT-mediated reprogramming of the TME is expected to improve the efficacy of immune checkpoint blockade in immune-cold carcinomas and to provide new therapeutic avenues for fatal diseases. However, further research is needed to fully realize the potential of implementing such strategies.

## 6. Conclusions

This review summarizes the interaction of elevated polyamine levels in tumor cells and immune cells and analyzes the relevance of their supportive role in carcinogenesis and the link between energy metabolism and TME, leading to the use of polyamines as anticancer interventions. Drugs or polyamine analogs targeting polyamines and key enzymes associated with polyamine metabolism are effective against cancer in experimental animal models, and some of them have been evaluated in clinical trials. However, there are still some problematic areas that need to be addressed, and the specific mechanisms by which polyamine metabolism affects HCC are still unclear; in this paper, we propose mitochondrial autophagy as a possible mechanism by which polyamine metabolism affects HCC. However, how BNIP3L-mediated mitochondrial autophagy is regulated at the molecular level is not well studied, although it has been shown that the Beclin 1 autophagy gene plays an important role in cell growth control and tumor suppression in vivo, and Beclin 1 heterozygous deletion of mice exhibited accelerated development of hepatitis B virus-induced HCC [126]. Decreased Beclin 1 expression correlated with HCC grading, suggesting that Beclin 1 may be used as a prognostic biomarker for HCC [127]. This appears to be in conflict with our conjecture, but the role of Beclin 1 as a multifunctional protein is complex, and we hope to further confirm the roles played by Beclin 1 and mitochondrial autophagy in HCC through subsequent studies.

The relative increased dependence of tumor cells on polyamines, as well as the critical physiological role of polyamines in various immune cell types, makes targeting polyamine metabolic pathways a viable therapeutic strategy. Although inhibitors have been designed for all polyamine biosynthetic enzymes, the most successful inhibitor to date has been DFMO. DFMO irreversibly inhibits ODC through covalent binding to its active site, often leading to cellular arrest through depletion of putrescine and SPD [28,85,86]. Most importantly, these inhibitors have not provided satisfactory clinical results due to side effects and toxicity. The effectiveness of polyamine inhibitors in combination with other types of anticancer drugs has not been fully evaluated in clinical trials. Thus, despite considerable progress in innovative polyamine analogs and other polyamine-targeted agents, the development of effective and safe therapeutic agents needs to be further explored. In addition, the mechanisms of polyamine transport and catabolism in cancer cells are not fully understood. Therefore, a more detailed study of polyamine metabolism in tumorigenesis may contribute to the development of new safe polyamine inhibitors with promising results in the future.

## Figures and Tables

**Figure 1 ijms-25-00996-f001:**
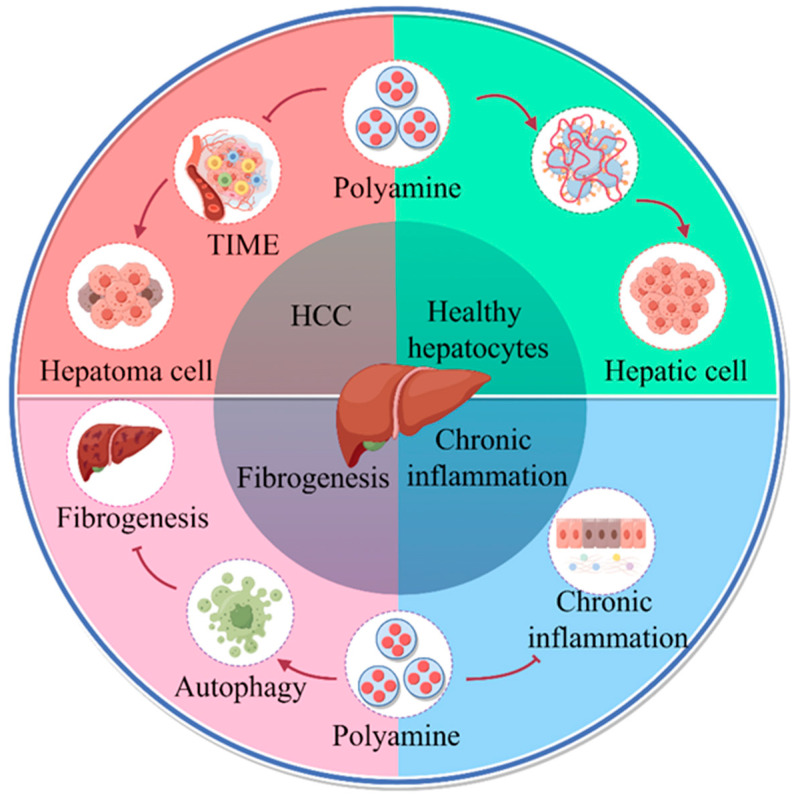
Role of polyamine metabolism in liver disease progression. Increased polyamine metabolism in cancer cells promotes the formation of an immunosuppressive microenvironment that facilitates liver cancer progression. Polyamines regulate replication, transcription, and cell division to promote hepatocyte growth, alter protein acetylation to regulate autophagy, and promote longevity. Polyamine metabolism inhibits hepatitis progression. Oral administration of SPD in liver fibrosis increases microtubule associated protein 1S (MAP1S) levels and activates autophagy to suppress oxidative stress and inhibit liver fibrosis (by Figdraw).

**Figure 2 ijms-25-00996-f002:**
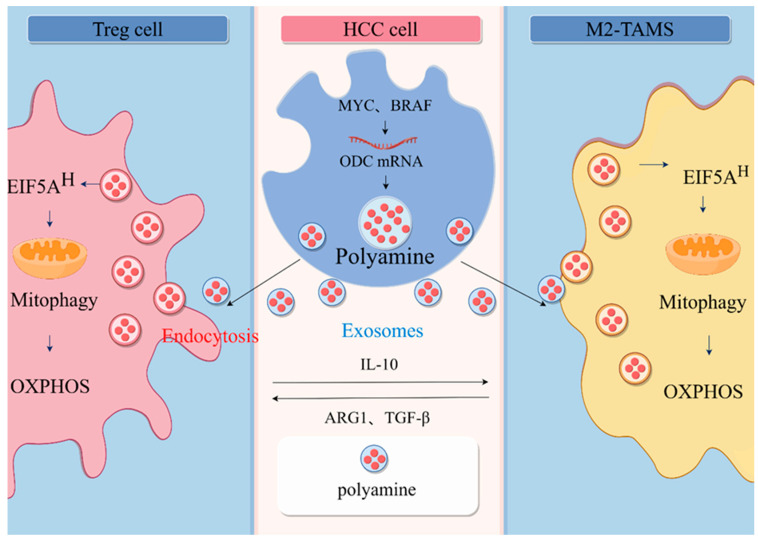
Communication between HCC and immune cells. In HCC cells, many oncogenes, including *MYC* and *BRAF*, contribute to the increased levels of polyamines in cancer cells. Polyamines are transported between cells in the form of vesicles, and immune cells can receive polyamines from tumor cell-derived extravasated vesicles, causing a rise in the polyamine pool of the immune cells, which, in turn, affects the immune phenotype of their immune cells. Elevated SPD enriches eIF5A^H^ in immune cells, leading to increased levels of eIF5A^H^-mediated translation of the OXPHOS complex and ultimately a metabolic shift to aerobic respiration, driving M2-type macrophage polarization and T cells differentiation to Treg cell (by Figdraw).

**Table 1 ijms-25-00996-t001:** Inhibitors of polyamine biosynthesis.

Drug	Target	Structure	Status
DFMO	ODC	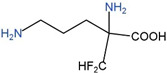	Approved for treatment of human African trypanosomiasis and hirsutism; multiple ongoing cancer clinical trials, including chemoprevention trials [89,90,91,92,93,94,95]
MGBG	AdoMetDC	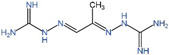	Phases 1 and 2 completed [96,97,98,99]
SAM486A	AdoMetDC	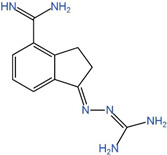	Phases 1 and 2 completed [100,101,102]
AdoDATO	SRM	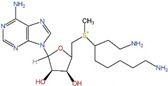	Preclinical use [103,104]
AdoDATAD	SMS	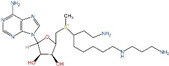	Preclinical use [105,106]

DFMO: α-difluoromethylornithine; MGBG: methylglyoxal(bis)guanylhydrazone; SAM486A: CGP48,664; AdoDATO:S-adenosyl-3-thio-1,8-diaminooctane; AdoDATAD:S-adenosyl-1,12-diamino-3-thio9-azadodecane; ODC: ornithine decarboxylase. AdoMetDC: adenosylmethionine decarboxylase; SRM: spermidine synthase; SMS: spermine synthase.

**Table 2 ijms-25-00996-t002:** Polyamine analogs.

Drug	Target	Structure	Status
BENSpm	ODC, AdoMetDC, SRM, SMS, SSAT, and/or SMOX	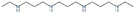	Phases 1 and 2 completed. Formulated into nanoparticles [108,109,110,111,112]
CPENSpm	ODC, AdoMetDC, SRM, SMS, SSAT, and/or SMOX	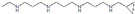	Preclinical use [113]
PG-11047	ODC, AdoMetDC, SRM, SMS, SSAT, and/or SMOX	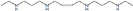	Phases 1, 1b and 2 completed. Formulated into nanopartides [114,115,116,117]
MDL72527	PAOX and SMOX	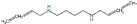	Preclinical use [118]

AdoMetDC: S-adenosylmethionine decarboxylase; ODC: ornithine decarboxylase; PAOX: acetylpolyamine oxidase; SMOX: spermine oxidase; BENSpm: N1,N11-bis(ethyl)norspermine;.

**Table 3 ijms-25-00996-t003:** Polyamine transport inhibitor.

Drug	Target	Structure	Status
Trimer44NMe	Polyamine transport	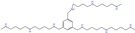	Preclinical use [41,120,121]
AMXT 1501	Polyamine transport	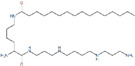	Preclinical use [119,122,123], Phase 1 trial in combination with DFMO completed (NCT03536728)
MeN44Nap44NMe (AP)	Polyamine transport	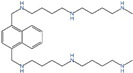	Preclinical use [124,125]

## Data Availability

Not applicable.

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
