# Peer review of "Polyamine Signal through HCC Microenvironment: A Key Regulator of Mitochondrial Preservation and Turnover in TAMs"

_ijms, 2024, doi:10.3390/ijms25020996_

Round 1

Reviewer 1 Report

Comments and Suggestions for Authors

Dear authors

The current review summarizes the current research progress of key enzymes and substrates of the polyamine metabolic pathway in regulating TAMs and T cells. You proposed that polyamine biosynthesis can intervene in the process of mitochondrial energy metabolism by affecting mitochondrial autophagy, which in turn regulates macrophage polarization and T-cell differentiation. You suggest that polyamine metabolism could be a key target for the interactive dialog between HCC cells and immune cells such as TAMs, so interfering with polyamine metabolism may become an important entry point to break the intercellular communication, providing new research space for developing polyamine metabolism-based therapy for HCC. This point is very interesting and covers many points that may help researchers in their future investigations.

Only few comments in the attached file

Best regards,

Reviewer 2 Report

Comments and Suggestions for Authors

Manuscript No ijms-2786132

„Polyamine signal through HCC microenvironment: A key regulator of mitochondrial preservation and turnover in TAMs” for International Journal of Molecular Sciences

Comments:

1. Introduction. Please also mention the role of IDO in regulating and limiting the activity of T lymphocytes. Moreover, it is worth mentioning the role of polyamine in the induction of IDO expression.

2. I ask the Authors to review the text again and limit the repetition of the same information in different subparagraphs.
